# Optical Mapping of Cardiomyocytes in Monolayer Derived from Induced Pluripotent Stem Cells

**DOI:** 10.3390/cells12172168

**Published:** 2023-08-29

**Authors:** Mohammed Djemai, Michael Cupelli, Mohamed Boutjdir, Mohamed Chahine

**Affiliations:** 1CERVO Brain Research Center, Institut Universitaire en Santé Mentale de Québec, Quebec City, QC G1J 2G3, Canada; 2Cardiovascular Research Program, VA New York Harbor Healthcare System, New York, NY 11209, USA; 3Department of Medicine, Cell Biology and Pharmacology, State University of New York Downstate Health Sciences University, New York, NY 11203, USA; 4Department of Medicine, NYU School of Medicine, New York, NY 10016, USA; 5Department of Medicine, Faculty of Medicine, Université Laval, Quebec City, QC G1V 0A6, Canada

**Keywords:** optical mapping, cardiac electrophysiology, membrane potentials, Ca^2+^ imaging, conduction velocity, iPS cells, hiPSC-CM monolayer, simultaneous dual optical mapping

## Abstract

Optical mapping is a powerful imaging technique widely adopted to measure membrane potential changes and intracellular Ca^2+^ variations in excitable tissues using voltage-sensitive dyes and Ca^2+^ indicators, respectively. This powerful tool has rapidly become indispensable in the field of cardiac electrophysiology for studying depolarization wave propagation, estimating the conduction velocity of electrical impulses, and measuring Ca^2+^ dynamics in cardiac cells and tissues. In addition, mapping these electrophysiological parameters is important for understanding cardiac arrhythmia mechanisms. In this review, we delve into the fundamentals of cardiac optical mapping technology and its applications when applied to hiPSC-derived cardiomyocytes and discuss related advantages and challenges. We also provide a detailed description of the processing and analysis of optical mapping data, which is a crucial step in the study of cardiac diseases and arrhythmia mechanisms for extracting and comparing relevant electrophysiological parameters.

## 1. Introduction

Fluorescence microscopy has been extensively used in biomedical research since the development of imaging techniques that use different optical elements and various fluorophores. Membrane voltage and Ca^2+^ signaling are vital physiological phenomena. In cardiac myocytes, action potentials (APs) are electrical signals that propagate and initiate electrical activity in neighboring cells. Intracellular Ca^2+^ homeostasis, including Ca^2+^ uptake and release by the sarcoplasmic reticulum and trans-sarcolemma influx/efflux, is integral to processes like AP generation and excitation–contraction coupling within the myocardium.

In cardiac electrophysiology, APs and Ca^2+^ transients (CaTs) have been measured and studied using traditional methodologies (e.g., patch clamps, multielectrode array systems for APs, and single cell imaging for CaTs) [1,2,3]. These techniques have enabled significant investigations into fundamental electrophysiological mechanisms that underlie both physiological and pathological conditions. Nevertheless, these established approaches have certain limitations, notably in terms of spatial resolution. This restriction becomes particularly challenging when studying excitation dynamics, as well as the genesis, sustenance, and termination of complex arrhythmia. Thus, the emergence of optical imaging techniques has allowed electrophysiologists to overcome these limitations and to study the propagation of AP and CaT waves using fluorescent probes. Numerous voltage-sensitive dyes (VSDs), featuring distinct emission wavelengths, have since been developed and subsequently refined, including the ANEPPS family (aminonaphthylethenylpyridinium), RH237, PGH-1, and FluoVolt [4,5,6,7]. Similarly, several fluorescent Ca^2+^ indicators (CIs) have been constructed, including Fura-2 AM, Fura Red, Indo-1 AM, Rhod-2 AM, and Fluo-4 [8,9,10,11,12]. The recent emergence of genetically encoded Ca^2+^ indicators (such as Cameleons, GCaMP, and Twitch) and voltage indicators (like FlaSh, VSFP, ArcLight, and Voltron) has provided novel tools for the field of cardiac optocardiography [13]. These indicators have nearly replaced organic probes in several applications, including in vivo imaging, high-throughput drug screening, and optogenetic studies [14,15,16]. However, organic fluorophores remain the tool of choice for in vitro studies, as they are more advantageous in terms of cost, ease of use, photostability, and signal-to-noise ratio.

The optical mapping of monolayers has provided valuable insight into the physiological and pathological mechanisms, including AP initiation and conduction, unidirectional conduction block, gap junction uncoupling, ischemia, alternans, and anisotropy [17]. Clinically, the measurement of conduction velocity (CV) via intracardiac electrogram represents a fundamental parameter in cardiac electrophysiology, enabling the quantification of the speed and direction of electrical signal propagation within the heart. CV measurement provides a valuable description of AP propagation dynamics, allowing for the identification of arrhythmic patterns, precise localization of pathological substrates, and the stratification of patient vulnerability to high-risk adverse cardiac events [18].

The development of human-induced pluripotent stem cell (hiPSC) technology [19,20], along with cardiomyocyte-oriented differentiation methods [21,22,23], has opened new horizons in the field of cardiac electrophysiology. In recent years, the cultured hiPSC-derived cardiomyocyte (hiPSC-CM) monolayer has become a contemporary in vitro model for the study of anisotropic conduction and arrhythmogenesis at the tissue level. This two-dimensional cardiac model is very advantageous in terms of cost-effectiveness, efficient high-throughput capabilities, technical ease, and a multitude of standardized plate-based applications [24]. Optical Ca^2+^ and voltage mapping in hiPSC-CM monolayers has become a powerful and well-established tool for cardiac disease modeling over the last decade [25,26,27].

One limitation of the hiPSC-CM monolayer is its predominant composition of heterogeneous and immature cardiomyocytes, which may not fully represent the complexities of the adult human heart [28]. Additionally, this model lacks the ability to faithfully replicate the three-dimensional structure and orientation of the myocardium, which influence the heart’s electrical and mechanical properties [29]. Despite these limitations, hiPSC-CM monolayers offer distinctive advantages over unicellular models by forming a functional syncytium that can conduct an excitation wave [30]. Thus, this two-dimensional cardiac tissue serves as a valuable intermediary between unicellular and whole-organ models.

## 2. Cardiac Optical Mapping

### 2.1. Optocardiography

Optical mapping of the excitation dynamics and rhythmic activation of the heart was introduced at the beginning of the last century [31]. Contractions of perfused beating frog hearts were recorded for the first time in 1916 at 15 frames per second (FPS) on bromide paper using a cinematographic apparatus (Figure 1a) by the cardiac electrophysiologist G. Mines [32]. Mines was a pioneer contributor to cardiac electrophysiology because of his research into arrhythmias such as tachycardia and fibrillation. His findings serve as the basis for our current understanding of these phenomena [33]. Later, Carl J. Wiggers used the same cinematographic apparatus to monitor irregular contraction during ventricular fibrillation [34]. Interestingly, cardiac electrophysiologists today still rely on the advantages of the optical measurements developed by Mines and Wiggers and use cinematography in high spatial resolution studies [31].

Initially, the mapping of cardiac electrical activity was primarily performed using properties of light, such as scattering and birefringence, to record activation in different regions of the myocardium. The advancement of technology in the 1960s and 1970s led to the development of novel tools for studying cardiovascular diseases in Langendorff preparations. In 1976, G. Salama and M. Morad used a potentiometric probe (merocyanine 540) to stain frog hearts and measure changes in transmembrane potentials (Figure 1b). They concluded that the recorded fluorometric variations (1.5 to 2%) and APs measured using intracellular microelectrodes were similar in various regions of the heart (pacemaker and atrial and ventricular tissues) [35]. However, the VSDs that existed at the time, in particular merocyanine 540, were very limited in terms of signal quality and response speed, which led to the development of VSDs, such as oxonol and styryl dyes [4,36]. As the responses of these fluorescent probes vary from one preparation to another, several analogs of these fluorophores were developed to obtain a brighter signal with the least noise possible [37].

Since the 1990s, several cardiac electrophysiology laboratories have begun using optical mapping of membrane electrical activity for their research [38,39,40,41]. Moreover, several VSD probes have been used to study changes in membrane potentials in several excitable tissues (Figure 1c). A wide variety of faster (temporal resolution of a few microseconds), brighter, and less toxic VSDs are now available. They are characterized by their color (excitation/emission), fluorescence lifetime, response speed, intensity, and signal-to-noise ratio.

**Figure 1 cells-12-02168-f001:**
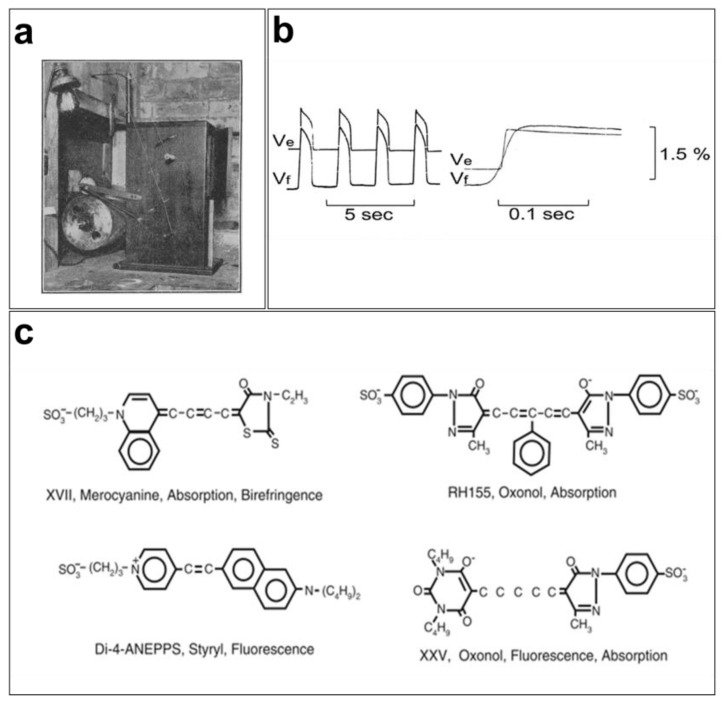
**Early cardiac optical mapping**: (**a**) first cinematographic apparatus used by G. Mines to map and record contractions of perfused frog hearts [32]; (**b**) simultaneous AP recordings from frog hearts using the potentiometric dye merocyanine 540 (Vf) and microelectrodes (Ve) produce faithful recordings of transmembrane potentials using fluorescent signals [42]; (**c**) merocyanine 540, RH155, Di-4-ANEPPS, and XXV are some of the fluorophores that have been used to record membrane action potentials [43].

### 2.2. VSDs and Ca^2+^ Indicators

A basic knowledge of the chemistry and physics of the VSDs and fluorescent Ca^2+^ indicators is required to optimize cardiac optical mapping experiments. Both types of probes have different physicochemical properties in terms of the absorption/emission spectra and mechanisms for sensing and reporting parameter variations. These dyes are often combined to simultaneously measure APs and CaTs.

#### 2.2.1. Voltage-Sensitive Dyes

Fluorophores are chemical species that can re-emit photons after being excited by ultraviolet or visible electromagnetic radiation. They have different properties that distinguish them from each other, such as fluorescence lifetime, quantum efficiency (the number of photons re-emitted relative to those absorbed), and Stokes shift (the difference in nanometers between the peaks of the absorption and emission spectra for the same transition). Each fluorescent molecule has several electronic states. When these molecules are exposed to a photon with an energy *hν* (where *h* is the Planck’s constant and *ν* is the frequency), precisely matching the energy gap between the ground state and the excited state, the photon can be absorbed. This absorption excites the fluorophore which then loses this energy after a few nanoseconds through either vibrational relaxation or fluorescence emission [44].

VSDs are organic molecules developed to report membrane potential variations in excitable cells and tissues by changing their optical proprieties (absorbance, fluorescence, or birefringence) [45]. Like fluorophores, VSD molecules contain a specific portion, called the chromophore, which senses the light and re-emits photons at a certain energy (wavelengths). Efficient VSDs usually have chromophores that are bright, photostable, and highly sensitive to the membrane electrical field. Therefore, the choice of the proper VSD is crucial for experimental efficiency. Many fluorescent potentiometric probes are commercially available and are classified into two types: slow or fast response. Potentiometric dyes, such as ANEPPS and RH237, interact with the surrounding electric fields, which change their intramolecular charge distribution resulting in a very fast change in their spectral properties. This fast optical response is proportional to the membrane potential changes, which makes it possible to measure transient APs in the millisecond range. In contrast, VSDs, including merocyanine and oxonol, sense the membrane voltage changes with a much slower mechanism described below [36].

VSDs transduce APs by interacting with the cell membrane, which changes the VSDs’ electronic configuration via various mechanisms. First, dyes containing cyanine and oxonol chromophores use the ON–OFF switch mechanism [46], which consists of bonding the positively charged dyes to the membrane when the resting potential is negative and releasing the dyes during the depolarization phase. This mechanism makes fluorescent dyes more sensitive to changes in the membrane potential but results in a slow response time. The fluorescence resonance energy transfer (FRET) mechanism, on the other hand, relies on an energy transfer between a donor chromophore that is attached to the membrane’s extracellular surface and a membrane permeant acceptor chromophore that is negatively charged. At the resting potential, the acceptor is located on the outer membrane’s surface, enabling the energy transfer and, thus, emitting fluorescence. During depolarization, the acceptor will penetrate the membrane down to the intracellular side, which reduces the fluorescence emission wavelength [47]. This ratiometric voltage-sensing mechanism is very fast and sensitive. However, it requires the application of two probes to the preparation. The reorientation mechanism is another way dyes can interact with the membrane electrical field, that is, by changing their orientation [48].

Electrochromism is a property of some chemical species that allows them to reversibly change their color and opacity in the presence of a surrounding electric field. A direct interaction between an electrochromic fluorophore and the membrane potential will cause the Stark effect (Figure 2), which leads to a small shift (a few nanometers) of the excitation or emission spectra to a shorter wavelength (hypsochromic shift) or to a longer wavelength (bathochromic shift) in response to variations in the membrane potential. Members of the ANEPPS family are among the most widely used electrochromic VSDs for cardiac mapping. They include 1-(3-sulfonatopropyl)-4-(β-{2-(di-N-butylamino)-6-naphthyl} vinyl) pyridinium betaine, also known as di-4-ANEPPS. This probe was developed as a reliable and flexible alternative to the other dyes described above and is suitable for most experimental situations. Di-4-ANEPPS is an electrochromic VSD and has an ultrafast and linear response to changes in membrane voltage. In fact, it is fast enough to follow APs of a few milliseconds, with a ΔF/F of up to 10% for a gradient of 100 mV (theoretically) [49]. It is worth noting that photoinduced electron transfer molecules, such as FluoVolt [50], have become increasingly popular in recent years, as they do not develop the rapid phototoxicity that afflict the organic molecules of the ANEPPS family; however, the ANEPPS molecules are still the most commonly used.

Orbital calculations performed on the di-4-ANEPPS molecule have shown that, in the ground state, its positive charge is generally located near the pyridinium nitrogen, while in the excited state, the charge moves toward the aromatic amines (Figure 2a) [7]. Once in contact with a cell, the fluorophore easily intersperses among the lipid molecules of the membrane and is oriented perpendicular to its surface. When it is excited (change in polarity), the positive chromophore load becomes parallel to the orientation of the electric field in the membrane. When di-4-ANEPPS is excited and the membrane is at its resting potential, the polarity of the fluorophore changes and retransmits light with a wavelength of λ_0_. When APs occur, the emitted wavelength changes to λ_1_, and the difference λ_0_ − λ_1_ becomes proportional to the difference in the membrane potential [49].

The optical properties (absorption and emission) of a VSD depend significantly on the nature of the preparation, its thickness, and the recording medium. When cardiac tissue is loaded with di-4-ANEPPS, it is excited with a 530 nm green light and re-emits fluorescence with a peak emission at 620–670 nm (red light). When the membrane begins to depolarize, the emission spectrum of the fluorescent molecule undergoes a shift of a few nanometers to a shorter wavelength (Figure 2b,c), resulting in a decrease in fluorescence intensity. When the membrane repolarizes, the emission spectrum returns to its original state, and the fluorescence intensity increases again. This mechanism enables di-4-ANEPPS to make rapid and linear responses to changes in the transmembrane potential (Figure 2d). These two properties make this probe the tool of choice for many cardiac optical mapping applications.

Similarly, RH dyes (including the RH237) are electrochromic VSDs, which were primarily developed for neuronal imaging by Rina Hildesheim and colleagues [4]. These styryl dyes exhibit a spectral shift of their excitation and emission proprieties following a membrane potential change. In cardiac electrophysiology, RH237 dye has been used for the simultaneous mapping of APs and CaTs in Langendorff perfused hearts [51] and hiPSC-CM monolayers [52] because of its greater Stokes shift compared to di-4-ANEPPS (λ_Ex_/λ_Em_ = 528/715 nm in Tyrode’s solution). A longer emission wavelength reduces spectral overlap when a Ca^2+^ indicator as Rhod-2 AM is used at the same time for dual imaging experiments.

#### 2.2.2. Ca^2+^ Indicators

The regulation of calcium in cardiomyocytes plays a critical role in the heart, including the excitation–contraction coupling. This physiological mechanism is crucial for proper heart functioning [53]. During an AP, the opening of the L-type voltage-gated calcium channels allows for an inward influx of calcium into the cell. This initiates the release of calcium from the sarcoplasmic reticulum, which activates contraction [54]. The release of calcium is mediated by ryanodine receptors (RyR), which are triggered by localized subsarcolemmal calcium entry. This process is known as calcium-induced calcium release (CICR) [55] and underlies calcium sparks. These sparks are essentially rapid and transient increases in the calcium concentration within a small region of the cell. In pathological conditions, such as heart failure (HF), improper regulation of cellular Ca^2+^ levels can impact an AP’s duration, leading to a spontaneous membrane depolarization [56]. Dysregulation of intracellular Ca^2+^ in cardiomyocytes is a risk factor for contractile dysfunction and arrhythmogenesis in failing hearts [57]. Hence, the simultaneous measurement of APs and CaTs is crucial for our understanding of the mechanisms of HF and inherited Ca^2+^-mediated arrhythmias, such as catecholaminergic polymorphic ventricular tachycardia (CPVT) and cardiomyopathy.

Sydney Ringer was the first to suggest the importance of Ca^2+^ in the cardiac contraction mechanism in the early 1880s [58,59]. Since then, the role of CaTs in the initiation of excitation–contraction coupling within the heart has been widely studied [60]. However, direct measurements of CaTs were only possible a century after Ringer’s discovery (Figure 3a) [61,62]. Indeed, the introduction of the first generation of fluorescent CIs in the mid-1980s has greatly increased our knowledge of Ca^2+^ dynamics and cardiac arrhythmias [63]. Several CIs are currently available, including organic probes, which are the most widely used because of their bright signal and broad range of binding affinities to intracellular Ca^2+^. When perfused into cells, these indicators selectively bind with free cytoplasmic Ca^2+^ ions and shift their absorption or emission spectrum or change their emitted fluorescence intensity in response to variations in the Ca^2+^ concentration. In fact, chemical CIs can be divided into distinct groups based on their proprieties, e.g., ratiometric versus nonratiometric, high- or low-binding affinity (dissociation constant, Kd), and excitation/emission wavelengths. Hence, it is important to select the most suitable probe for a given experiment based on its spectral characteristics.

Choosing the right Ca^2+^ indicator requires careful consideration. While using different CIs might produce comparable results with proper calibration, they inherently act as Ca^2+^ buffers that affect physiological signaling. Finding a balance between the intensity of the CI’s signal and its concentration is crucial. Selecting a CI with lower Ca^2+^ affinity can minimize buffering effects but may also reduce the intensity of the signal. Furthermore, when dealing with multiple fluorophores or autofluorescence, considering the spectral properties of the indicators can help guide the choice [64]. Furthermore, commercially available CIs are provided in various chemical forms, such as salt (acid), acetoxymethyl ester (AM), and dextran conjugates. Each type of these fluorescent indicators requires a specific loading procedure depending on their nature. Salt and dextran CIs are membrane impermeable and must be introduced into cells using invasive techniques (microinjection, electroporation, or lipotransfer). Therefore, membrane permeable (i.e., hydrophobic) AM ester indicators are more suitable for excitable tissue imaging, as they can simply be added into the extracellular media for staining. When they passively permeate the cell membrane, the AM group is cleaved by the intracellular esterase, and the fluorescent CI molecules are trapped in the cytoplasm [64].

To ensure accurate measurements of CaTs, it is important to choose a CI with minimum perturbation of the Ca^2+^ dynamics. CaT imaging in cardiac cells and tissue, which undergo significant and rapid variation in [Ca^2+^]_i_, requires a bright and fast-responding indicator, preferably without spectral interference, with the potentiometric dye for dual imaging. Rhod-2 AM is a single-wavelength, rhodamine-based Ca^2+^ indicator (λ_Ex_/λ_Em_ = 532/585 nm in Tyrode’s solution) engineered by Tsien and co-workers in 1989 (Figure 3b) [10,65]. Because the excitation/emission range of Rhod-2 is long enough to reduce autofluorescence [65], this dye is often used for in situ CaT measurement in brain [66,67,68] and heart preparations [69,70,71]. Moreover, this indicator has also been used to measure mitochondrial Ca^2+^ levels, since it can sequester into this organelle because of its positively charged AM group [10,64,65].

## 3. Induced Pluripotent Stem Cells

The limited access to human cardiac tissue has always been a major challenge for the study and modeling of arrhythmia and other cardiac diseases. The introduction of human embryonic stem cells (hESCs) has opened many possibilities given that they serve as an unlimited source of cardiac cells and tissues. However, research with hESCs was met with a host of ethical and moral concerns regarding their acquisition and use [72]; thus, an alternative model was needed to circumvent these limitations. This became possible following the introduction of mouse iPSCs by Kazutoshi Takahashi and Shinya Yamanaka in 2006 [73]. These two Japanese researchers were able to reprogram already differentiated adult somatic cells (fibroblasts) to the pluripotent state using four transcription factors: Oct 3/4 (octamer-binding transcription factor 3/4), Sox2 (sex-determining region Y)-box 2, Klf4 (Krüppel-like factor 4), and c-Myc. Like ESCs, iPSCs have the potential to differentiate into any cell type while keeping the same genetic background [74]. In 2007, the same reprogramming method was used by Yamanaka et al. to generate human stem cells from adult fibroblasts (hiPSCs) [19]. At the same time, James Thomson et al. generated hiPSCs using other transcription factors [20]. Since then, several studies have followed in their footsteps and have improved the techniques for the differentiation and generation of hiPSCs. These cells are currently being utilized by numerous researchers owing to their potential for disease modeling and therapeutic applications [74,75].

The study of cardiac diseases and arrhythmias is often limited by the availability of affected tissues, as well as difficulties in maintaining primary patient cells in culture. These problems have been circumvented by the introduction of hiPSCs, where somatic cells from skin biopsies, blood samples, or more recently from lymphoblastoids [76] are reprogrammed to pluripotency and then differentiated into cardiomyocytes that retain the genetic characteristics of the patient. In addition, hiPSCs can be multiplied almost without restriction and are considered an unlimited source of any type of human cells. The aim of disease modeling using hiPSCs is to broaden our knowledge of the different pathogenic mechanisms involved in cardiac diseases and, more importantly, to develop potential treatments. hiPSCs have applications for cardiotoxicity assessments, which present a serious challenge for the medical treatment of cardiovascular diseases [77]. Therapeutic agents are usually tested on animal models [78]. However, the biology of rodents is fundamentally different from that of humans, which is one of the main causes of the failure of human clinical drug trials, with only 16% of drugs being approved after clinical trials [79]. As such, hiPSCs are an attractive, practical, and economically profitable tool for disease modeling and can be used for preclinical cardiotoxicity and pro-arrhythmic drug screening [80].

The effective differentiation of hiPSCs into functional contractile CMs has been achieved by modulating several signaling pathways that are associated with heart development during the embryonic stage, such as Wnt and Notch [81,82,83]. Several differentiation protocols have been developed since the introduction of hiPSCs, among which monolayer-based differentiation stands as an efficient strategy for creating highly pure populations of CMs. Many commercialized hiPSC-CM differentiation kits are currently available providing more than 80% efficiency and mixed CMs subtypes, but the majority have a ventricular-like phenotype. Advances in stem cell research have enabled the ability to orient the in vitro differentiation process toward a specific CM subtype including atrial-derived hiPSC-CMs [84]. This has opened new horizons in cardiac electrophysiology, chamber-specific disease modeling, targeted therapy, and pharmacology [85,86].

### Cultured hiPSC-CM Monolayer System

CiPA (comprehensive in vitro proarrhythmia assay) is an innovative nonclinical approach to evaluate the electrophysiological proarrhythmic mechanisms, cardiotoxicity, and safety of potential drug candidates [87]. This new assay is being developed by a coordinating collaborative effort of academic research laboratories, federal regulatory agencies, and the pharmaceutical industry [88]. The central components of CiPA are hiPSC-CMs, which offer a human-based model for pre-clinical cardiotoxicity testing [89]. The use of hiPSC-CMs provides several key benefits, such as the availability of unlimited quantities from commercial sources, cryopreservation, and the presence of a diverse array of ion channels, providing a more integrated system compared to heterologous cell systems that only test a single ion channel [89,90].

hiPSC-CMs have been widely employed in cardiac disease modeling and drug screening studies. However, these studies have mostly been conducted on isolated single cell preparations, which may not accurately represent the native cardiac tissue phenotype [91]. Recent advancements have allowed for the generation of 2D monolayer, 3D cardiac spheroids and organoids and engineered cardiac tissues (EHTs) from hiPSC-CMs [92,93]. These structures offer a closer representation of native cardiac tissue and can provide more accurate results in disease modeling and drug screening studies [92,94]. One significant advantage of hiPSC-CM culture models is the ability to form electrically coupled syncytia, which can be used to study the propagation of electrical impulses and mechanisms of re-entrant arrhythmias. Thus, hiPSC-CMs provide a valuable tool for investigating and understanding the complexities of cardiac disease as well as for high-throughput cardiotoxicity screening [28,95].

Generating 2D hiPSC-CM monolayers is usually achieved by plating the CMs on a flat adherent surface often coated with a basement membrane, such as Matrigel^®^ or Geltrex^TM^ [96]. This 2D hiPSC-CM monolayer model offers several advantages over 3D models, such as increased accessibility for optical mapping techniques and better visualization of cellular and subcellular structures. Additionally, 2D hiPSC-CM monolayers are easier and less expensive to maintain, and they allow for high-throughput screening and analysis [92]. However, it is important to note that 2D hiPSC-CM monolayers do not fully capture the complexity and physiological behavior of the heart, so they should be used in conjunction with other models to obtain a comprehensive understanding of cardiac function [97].

## 4. State-of-the-Art hiPSC-CM Monolayer Optical Mapping

### 4.1. Design of the Optical Mapping System

An optical mapping system is mostly composed of three essential elements: a high-speed optical detector, a light source, and a macroscope configuration setup (including filters and lenses) [98]. When a hiPSC-CM monolayer is stained with a VSD and/or a CI, a rapid propagation of AP/CaT produces small (low-signal) changes in fluorescence intensity over the background [ΔF/F_0_]; thus, a high-speed, low-noise photodetector is required. It is important to note that the relationship among the spatial resolution, temporal resolution, and signal-to-noise ratio (SNR) constitutes one of the fundamental limitations governing any optical detection system [99]. Increasing the spatial resolution by adding more pixels to a fixed area may not necessarily increase the SNR, as the fixed number of emission photons will be spread over more pixels. On the other hand, increasing the temporal resolution may result in a reduced optical signal, as there will be less time to collect the same number of photons. Therefore, achieving a balance among these parameters becomes crucial when choosing an appropriate detector based on the application for the optimal light detection.

Furthermore, image magnification is essential in many applications, especially in biomedical imaging. In fact, optical microscopes have been designed and optimized to be able to observe very small specimens at close distances using light with wavelengths ranging from UV to IR (ultraviolet to infrared). Conventional microscopes using high-magnification objectives (from 4× to 200×) efficiently collect light, providing very clear and bright images with good contrast, thanks to their large numerical apertures (NAs) [100]. In fact, the light collection efficiency of a microscope is proportional to the NA^4^ of its objective. However, these high-magnification microscopes with large NAs do not allow for imaging over a large field of view. Moreover, commercial photography objectives have often been used in several applications of macroscopic biomedical imaging, since they allow for a wider field of view (up to a few tens of millimeters) and a large working distance while having almost the same light collection efficiency as that of high-magnification microscopy objectives [39,99,101].

Every optical imaging system is fundamentally composed of one or more lenses. A lens is a transmissive optical device that focuses or disperses a light beam because of its property of refraction. In addition to the illuminance (i.e., irradiance), several factors can determine the brightness of the image, such as the field of view, refractive index of the sample (and the imaging media), diameter and NA of the lens constituting the objective, as well as the distance between the objective and the imaged sample. Like conventional fluorescence microscopes, an optical mapping macroscope is composed of several optical elements (emission and excitation filters, objectives, collimating lenses, and dichroic mirrors) in addition to a light source and a photodetector. Currently, numerous types of optical detectors are used depending on the application, as well as the nature of the samples to image. Among them are photomultiplier tubes (PMTs), known for their high sensitivity, photodiode arrays (PDAs), CCD cameras (charge-coupled device), and CMOS cameras (complementary metal oxide semi-conductor). The core element in every optical mapping setup is the photodetector. Therefore, the selection of an appropriate detector must be based on experimental parameters, as well as the fluorophores used. Moreover, when imaging electrical or calcium activity in cardiomyocytes or cardiac tissue, the chosen sensor must have the following characteristics: high sensitivity with minimum background noise; temporal resolution > 500 Hz; spatial resolution with pixels < 100 µm, and pixel density >$\;1\times 10^{6}/{\mathrm{cm}}^{2}$. The pixel size should be in the range of tens of microns to balance the high spatial resolution with a sufficient SNR. With very small pixels (<3 µm), one can achieve very high spatial resolutions at the expense of the SNR, while with very large pixels (>100 µm), one loses spatial information [41,99,102].

CCD cameras have been the detector of choice for several years, since they allow for imaging a large field of view thanks to the high pixel count that their sensors (active surface of CCD) have, with a high temporal resolution, quantum yield (Φ), and SNR. This makes CCDs very efficient, especially in very low light conditions [42]. However, these sensors are limited by their acquisition speed, which makes fast events difficult to image, such as the depolarization rate of a cardiac AP, which occurs within the range of milliseconds. On the other hand, CMOS cameras can achieve ultrafast acquisition speeds (up to 10 kHz) while keeping the advantages of CCDs (high Φ and SNR), which makes them more attractive for researchers despite their higher cost [98].

Several types of light sources have been used to optically map AP and CaT propagation in cardiac tissue, such as lasers, halogen incandescent lamps, mercury lamps, xenon discharge lamps, and light-emitting diodes (LEDs) [41]. Lasers are appropriate for imaging systems using PMT or PDA because of their very high irradiance. However, LEDs are more advantageous for high-speed imaging, since they are rapidly switchable and generate intense, stable monochromatic light compared to other sources of illumination, apart from lasers.

For macroscopic optical mapping, three main configurations can be used, including lensless, single lens, and tandem-lens systems (Figure 4) Despite their differences, advantages, and limitations, these optical configurations offer a viable light collection efficiency at low magnification [99]. The lensless system consists of directly projecting the image on the active surface of the photodetector. The latter could be very close to the sample through a glass coverslip or related to an optical fiber to avoid the potential risks of damage, as in the “contact fluorescent imaging” (CFI) system. A custom-designed CFI setup, developed by Entcheva et al., in the early 2000s, was successfully used to optically map transmembrane potentials and re-entry in monolayers of cultured neonatal rat ventricular myocytes (NRVMs) [103,104]. A CFI system offers some advantages over conventional lens-based microscopy and has some features that make it suitable for imaging cultured cell monolayers. This optical configuration not only exploits the planar geometry of the cardiac monolayer, so it does not require any focusing mechanism, it also avoids some of the optical distortions and reflections that occur at the interface between glass and air thanks to the uniformity of transmitted light through the optical fibers. It also provides a fixed magnification independently from the working distance (WD) (Figure 4a). However, the CFI system also presents certain limitations, including its reliance on transillumination excitation and the restricted ability to adjust spatial resolution or magnification [99].

A simpler optical design is to use a single lens with a high NA and low magnification in front of the photodetector (Figure 4b). This approach offers a useful WD and light collection efficacy. However, it has some drawbacks. For example, lenses with high NA and low magnification tend to have a significant vignetting effect, which means that the light intensity decreases as the distance from the optical axis increases. They also introduce spherical aberrations when they are used at short distances from the sample. Moreover, these lenses only achieve their maximum NA when they are focused at infinity. When they are used at finite WDs to the object (usually a few centimeters), the effective NA is lower [99].

Finally, the tandem-lens system is the most popular approach for cardiac optical mapping [105]. A tandem-lens system is an epi-fluorescence configuration (identical excitation/emission light pathway) composed of two fast lenses (L1 and L2) focused to infinity and facing each other [106]. The L1 will act as the objective lens, which collects light from the sample and forms an image at its focal plane, and the L2 will act as the eyepiece lens, which magnifies the image formed by the objective lens and projects it to the photodetector (Figure 4c). Thus, the magnification of a tandem-lens system depends on the focal lengths of the two lenses. This configuration enables faster shutter speeds and allows for efficient light detection with a practical WD. However, sophisticated components are often used to avoid optical aberrations and undesirable reflections [99,106,107].

### 4.2. Simultaneous Dual Optical Mapping

In addition to all of the different components and configurations described above, the complexity of an optical mapping setup also depends on the parameters of interest. The monitoring of AP has provided us with a basic understanding of electrical heart activity, sinus rhythm, and arrhythmogenesis, but many other potentially pathogenic parameters can be optically mapped. To further study physiological heart function, new fluorescent dyes were developed to measure ionic homeostasis (calcium, sodium, pH, etc.), redox state (NAD+/NADH), and oxygen levels. In addition to monitoring all of these physiological parameters separately, more complex optical mapping systems were designed to simultaneously combine them for deeper investigations into the mechanisms of cardiac arrhythmia.

The complex coupling between the electrical activity and intracellular Ca^2+^ dynamics is critical for proper cardiac function, thus highlighting the importance of the simultaneous multiparameter imaging [108]. B. Choi and G. Salama were the first to monitor APs and CaTs on perfused guinea pig hearts simultaneously. This was possible using two photodetectors (Figure 5) to collect the emitted fluorescence signal from the VSD and CI (RH237 and Rhod2, respectively) [109]. A few years later, a similar design was used by Fast and Ideker to map AP and CaT in a cultured cardiomyocyte monolayer [110]. Subsequently, technological advances (photosensors, LED light, and fluorescent probes) have allowed for the optical mapping of AP and CaT propagation in hiPSC-CM monolayers at the same time with a fairly high spatiotemporal resolution [52,91,108]. Assembling a dual optical mapping system can be costly and technically challenging [11,99]. It also requires the proper alignment of the sensors, filters, dichroic mirrors, light source, and lenses to ensure an overlapping signal from the same region of interest (ROI). Simultaneous optical mapping can also be performed using a single photodetector setup [111]. Single senor, multiparametric mapping uses a fast sequential excitation light switching and patterning to image each complementary dye (VSD and CI) [112], resulting in a substantial decrease in the temporal resolution proportional to the number of wavelength channels.

The simultaneous optical mapping of the voltage and calcium parameters necessitates the use of a VSD and CI, both introduced into the cardiac tissue and excited by either the same or distinct wavelengths. After being gathered using a lens and divided into separate optical channels using a dichroic mirror, the emitted fluorescence signals are then filtered and directed to the photodetectors (Figure 5). Many dual probe combinations have been developed and used in cardiac simultaneous optical mapping (Di-4-ANEPPS/Indo-1 [113], RH237/Fluo-3 AM [114], and di-8-ANEPPS/Fura-4F [52]), although the RH237/Rhod2-AM combination is still commonly used, since it requires a single light source (green: 530 nm) and offers a sufficient SNR with minimal spectral overlap [111]. The recommended optical macroscope configuration for the dual imaging is usually a tandem lens system (Figure 5), since it not only allows for the addition of a second photodetector because of the infinity-corrected light path, but it also offers a significant reduction of photobleaching and phototoxicity (because of the increased fluorescence collection efficiency, less intense excitation light is needed), increased SNR, and a brighter fluorescent signal [106,107].

### 4.3. Experimental Procedure

In general, the optical mapping of a hiPSC-CM monolayer is performed within 30 and 60 days of cardiomyocytes maturation. Performing a media refresh for the cardiac tissues approximately an hour prior to acquisition is recommended to significantly improve the results. The preparations are then stained with a VSD and/or CI diluted in Tyrode’s solution (e.g., 154 mM NaCl, 5.6 mM KCl, 2 mM CaCl_2_, 1 mM MgCl_2_, 8 mM D-glucose, and 10 mM HEPES; pH 7.3). After incubation in a controlled environment (37 °C, 95% O_2_, and 5% CO_2_) for several minutes (depending on the dye), the hiPSC-CM monolayers are usually washed with Tyrode’s solution and incubated in humid atmosphere for an additional 10 to 30 min. The optical recordings are usually performed at physiological temperature (37 °C) maintained by a warming plate or a custom-made perfusion chamber. Then, optical AP and CaT propagation are mapped across hiPSC-CM monolayers at a high sampling rate (approximately 500 FPS) and full spatial resolution. First, spontaneous electrical and Ca^2+^ activities are recorded for comparison among the control, drug-treated, or pathogenic cell lines. The preparations are then electrically paced at different frequencies (0.5–3 Hz), as AP and Ca^2+^ parameters are known to be frequency dependent. This procedure is carried out to standardize the data and establish a suitable baseline for accurate comparison at a consistent, fixed rate. Square wave electric discharges (of 1–20 ms duration) are delivered to the CM monolayers using a pulse generator through unipolar or bipolar electrodes (usually platinum-iridium) [91,115]. The preparations could also be optically stimulated using optogenetic tools, such as channelrhodopsins [116]. These image acquisitions need to be conducted swiftly to prevent the evaporation of the medium.

Cardiomyocytes are the contractile units of the myocardium. Their contraction can generate motion artifacts in the fluorescence signal recorded during optical mapping experiments. This can distort the electrophysiological parameters, such as the AP durations (APDs). Blebbistatin is widely used in cardiac optical mapping experiments to remove these artifacts [117,118]. It is a well-characterized molecular myosin II inhibitor. Myosin II enables the contraction mechanism to act in concert with actin to form an actomyosin contracting network [119]. Blebbistatin is used as an uncoupler of the excitation–contraction mechanism in cardiac electrophysiology [120]. It has been shown that 5 to 10 μM blebbistatin does not affect electrical activity in rabbit heart tissue, including various ECG parameters, AP morphology, and refractory periods [120]. Blebbistatin at 10 µM has no effect on the optical AP recorded from hiPSC-CM monolayer, as shown in Figure 6.

### 4.4. Processing Recorded Optical Signals

During acquisition, optical mapping raw data are recorded as an intensity map. Most modern detectors are square and would have a resolution of *n* × *n* where *n* is the number of pixels per column and row. In this instance, recorded data are saved as a series of matrices (*n* × *n*), where each element represents a measure of fluorescence intensity at a particular time from a specific location on the surface of the hiPSC-CM monolayer. The data are then saved and viewed using custom-made or commercial software installed on a computer. The background image is usually obtained by averaging the fluorescence intensity of the first four frames post-recording. This background serves as a reference to better visualize optical signal changes over time. The reference can be chosen manually in the event of artifacts or AP/CaT restitution at the beginning of the electrical stimulation. However, this can be avoided by recording the background image separately if this is allowed for in the system. Raw optical signals can be displayed by selecting a point within the cardiac monolayer corresponding to the membrane voltage changes, after which digital filtering can be applied.

The most common method for reducing the noise of optical signals is spatial filtering [121]. If the value of each pixel at (*x*,*y*) coordinates at time t is defined as *D*(*t*,*x*,*y*), the *D* value can be replaced by the average of the neighboring pixels in the selected *P* × *P* pixel square by performing spatial filtering, knowing that the *P* parameter can be chosen (3, 5, 7, or 9) based on the noise in the recorded optical signal (Figure 7a). After selecting the region of interest, a 3 × 3 mean filter is applied to the entire matrix containing the recorded data (Figure 7b). Then, a *3* × *3* median filter is applied. The algorithm of this filter is similar to averaging, except that the selected pixel is replaced by the median value of 9 neighboring pixels within the *3* × *3* square. This filter further reduces noise and helps to eliminate spatial heterogeneity (Figure 7b). More elaborate filtering methods have been developed for optical mapping applications, including conical and adaptive spatiotemporal Gaussian filtering [122,123]. Additionally, temporal filtering can also be implemented for further noise reduction. This involves the use of a kernel to remove unwanted specific segments from the signal spectrum [121].

A baseline deviation is often observed in raw recorded optical signals (Figure 7c) because of photobleaching. This drift can distort data, especially in the calculation of activation times, as well as the measurement of CVs and APDs. Baseline drift can be adjusted using a polynomial or exponential regression, where the drift function is fitted to a polynomial of degree *n* or an exponential of the form *ab^x^* and subtracted from the raw signal (Figure 7c). The same process is used to filter CaT optical recordings.

### 4.5. Extracting and Measuring Electrophysiological Parameters

One of the advantages of optical mapping over other electrophysiology techniques is its ability to simultaneously record APs and CaTs across the heart tissue at a high spatial resolution. This makes it possible to generate activation maps by extracting the activation times (t_Act_) of each pixel during AP/CaT wave propagation through the cardiac monolayer [121]. The activation times of APs correspond to the maximum increase in the signal relative to time (dF/dtmax). In other words, it is the time at which the first derivative of the recorded optical signal is maximal, whereas the CaT activation time corresponds to the half-upstroke time. t_Act_ is calculated in the imaged sample within a given time interval for a single propagation of an AP/CaT, and an average is calculated afterwards to generate an isochrone map with a corresponding color code (Figure 8a). Each color represents a time interval that includes multiple activation times. Isochrone activation maps provide insights into the heterogeneities of cardiac depolarization and intracellular Ca^2+^ propagation in the hiPSC-CM monolayer.

The cell-to-cell conduction of APs in the myocardium depends on the ease of the movement of ions among myocytes via gap junctions [78]. AP and CaT waves propagate at a given rate, referred to as the CV and Ca^2+^ wave propagation velocity (CaPV), and are very important electrophysiological parameters. A reduced CV facilitates reentry [124] and provides a ripe substrate for the initiation of both atrial fibrillation [125] and VT [126,127]. While AP and CaT normally follow each other very closely, there are situations preceding and during arrhythmia where discordance (uncoupling) can develop between those parameters [128]. Measuring cardiac CV and CaPV used to be a complicated task due to the low spatial resolution of standard electrophysiological methodologies (microelectrodes). Fortunately, optical mapping makes it possible to measure CV and CaPV with ease using the activation time of each pixel. CV/CaPV can easily be conceptualized into one of two distinct approaches: either the distance of propagation is measured over a predetermined time or the time of propagation is measured over a predetermined distance [129], the latter of which is shown in Figure 8b. Using this approach, the estimation of macroscopic CV/CaPV involves selecting two points (D_1_ and D_2_) at the two ends of the activation map (indicating the initiation and the termination of the activation wave, respectively) and measuring both the distance between these points and the difference in activation times. Point coordinates D_1_(*x*_1_,*y*_1_) and D_2_(*x*_2_,*y*_2_) are extracted, where (*x*,*y*) are the ordered pairs in pixels, and the distance is calculated using a simple Euclidian formula:$$d=\sqrt{\left[\left(x_{2}-x_{1}\right)2+\left(y_{2}-y_{1}\right)2\right]}$$
where *d* is the two-dimensional distance in pixels, which can be converted to µm using the following equation:$$Distance\;in\;µm\;per\;pixel=\frac{\;Object\;dimension\;in\;µm\;\;}{Image\;dimension\;in\;pixels\times Lens\;focal\;length\;in\;µm}$$

Measuring the CV/CaPV in cardiac monolayers has two major advantages over traditional 3D heart preparations: (1) the curvature of the cardiac surface introduces a significant source of error when calculating the distance between points [129]; (2) the anisotropic nature of propagation in cardiac tissue requires the calculation of both longitudinal and transverse CVs; however, the monolayer allows for homogenous propagation of activation wavefronts and avoids subsurface events [17].

Another advantage of the optical mapping of electrical activity in cardiomyocytes is the ability to study AP repolarization and CaT uptake. In fact, optical signals could be used to measure APD and CaT durations (TDs). As with activation times, APDs are measured from the optical signals recorded in each pixel [121]. APD_80_ and APD_50_ are defined as the durations of APs at 80% and 50% of their repolarization, respectively (Figure 8c). Similarly, TD_80_ and TD_50_ are also measured at 80% and 50% of Ca^2+^ uptake, respectively (Figure 8e) [130]. AP and CaT upstrokes could also be quantified as depolarization rise times (t_Rise_) and half activation to peak time (t_HA_), respectively. The ascending phases of each AP and the baseline of the signal, as well as the maximums corresponding to the peak maximum values, are identified (Figure 8d).

To ensure that slight baseline perturbations (noise) do not preclude rigorous analysis, depolarization times are measured between 10% and 90% of the AP maximum amplitudes (time (90%)–time (10%)) [121]. This measurement is very important in cardiac electrophysiology and is often performed in the presence of Na^+^ channel dysfunction or a change in their biophysical properties, given that these channels are directly involved in phase 0 of cardiac APs. The CaT upstrokes, on the other hand, correspond to the half activation to peak time (Figure 8e). This measurement represents the slower cytoplasmic Ca^2+^ release component actioned by RyR via the CICR process [79]. AP and CaT amplitudes are also estimated by normalizing the recorded fluorescence signals (ΔF/F_0_). Furthermore, the CaT uptake phase is measured by fitting the descending curve with an exponential function (Figure 8e), which estimates the decay time constant (decay τ) of CaT [130].

### 4.6. Challenges, Recent Advancements, and Future Directions for hiPSC Optical Mapping

Over the last decade, optical mapping of hiPSC-CM monolayers has provided valuable insights into physiological and pathological mechanisms of arrhythmias. However, optical recording of AP and CaT in hiPSC-CMs presents several challenges, including issues related to dye spectral characteristics, photobleaching/phototoxicity, quantum efficiency and acquisition rates, lenses, filters, motion artifacts, signal processing, and data extraction [91,101]. Cell culture can also be challenging when it comes to hiPSC-CMs differentiation protocols and monolayer generation, as well as the maturity and subtype heterogeneity of this 2D cardiac model [29,131].

The optical mapping space has experienced several advancements in recent years in terms of detectors (e.g., improved low-light SNR and falling cost of CMOS cameras) [98] and fluorescent probes (e.g., FluoVolt [6,50]). Such technological advancements have fueled new optical mapping applications, such as optogenetics [13,16,98] and high-throughput imaging [6,28]. In turn, these new applications have exciting implications for experimental paradigms, such as precision medicine [132], where a patient’s own cells are the experimental model for pathology and targeted therapies. Furthermore, high-throughput longitudinal screening will allow for pathological phenotypes and off-target effects or other complications from potential therapies to be more readily identified and addressed compared to similar paradigms using commercial or generic cellular models [133,134].

## 5. Conclusions

CaTs and APs are considered vital physiological signals. It is well known that intracellular Ca^2+^ is considered a secondary messenger, since it is involved in various biological processes. At the cardiac level, Ca^2+^ homeostasis regulates the mechanisms of contraction and conduction, as well as the rhythmicity of the heart. On the other hand, APs propagate electrical signals within the myocardium and coordinate the rhythmic contraction of the cardiac chambers. These are essential for the functioning of the myocardium. Optical mapping, therefore, is a valuable complementary technique to the traditional electrophysiological methods for studying these physiological signals. The recent developments made in the field of indicators engineering and fluorescence microscopy not only allowed for the study of more complex arrhythmia and cardiotoxicity but also greatly improved the signal-to-noise ratios and facilitated the simultaneous recording of both AP and CaT in vitro.

The introduction of hiPSC-CM technology has offered unique opportunities for cardiac regenerative medicine, disease modeling, and drug screening. In recent years, most studies have been focused on the cellular properties of hiPSC-CMs, namely, their AP and ionic current recordings. However, a multicellular model like the hiPSC-CM monolayer has unveiled the unique potential of hiPSC technology by facilitating the study of more complex electrophysiological phenomena (conduction and reentry) and evaluation of long-term tissue remodeling processes, phenotypes, and drug effects.

Despite their limitations, hiPSC-CM monolayers offer distinct advantages over unicellular models because of their capacity to replicate certain aspects of cell–cell interactions and tissue-like behavior. This model provides a platform for exploring microenvironmental factors and enable the observation of relevant morphological changes and phenotypic responses. Furthermore, the ease of visualization, suitability for high-throughput screening, and simplified experimental manipulation make this 2D model a valuable tool for studying cellular activities and responses. It is important to acknowledge, however, that hiPSC-CM monolayers may not fully replicate the complex 3D environment and interactions found in vivo. Mechanisms underlying fibrillation and defibrillation are most easily understood in simplified cardiac models. The cultured cell monolayer is one such experimental model that provides great versatility for basic studies of the formation, stability, and electrical termination of rotors (i.e., spiral waves). Thus, 2D monolayers serve as a highly valuable intermediary that bridges the gap between unicellular and whole-organ models [27,135,136]. In addition, culturing hiPSC-CMs on patterned surfaces has allowed for the explicit control of the tissue organization and improved morphological and functional maturity [137]. Patterning monolayers to mimic structural heterogeneity can promote triggered activity and reentrant patterns [17].

The human origin of the cardiac tissues is more relevant clinically compared to animal-derived cellular models. This is because of the significant interspecies differences in the expression of relevant cardiac ion channels. The ability to generate chamber-specific hiPSC-CM tissues (ventricular, atrial, and nodal CMs) in combination with the optical mapping technique have allowed for the further investigation of chamber-specific electrophysiological disorders underlying cardiac arrhythmia (atrial fibrillation, ventricular tachycardia, bradycardia, etc.). Additionally, this platform will help in the development of personalized and specific pharmacology treatment.

Optical mapping of hiPSC-CM monolayers has become a powerful tool for cardiac disease modeling and cardiotoxicity drug screening, yet it remains challenging. Recent advances are overcoming these challenges, including the development of novel fluorescent dyes, optogenetics, and enhancement of hiPSC-CM purity and maturity, as well as the utilization of modern computational approaches for signal processing and data extraction. Taken together, these advances are further enhancing and emphasizing the unique role that optical mapping plays in preclinical electrocardiology research.

## Figures and Tables

**Figure 2 cells-12-02168-f002:**
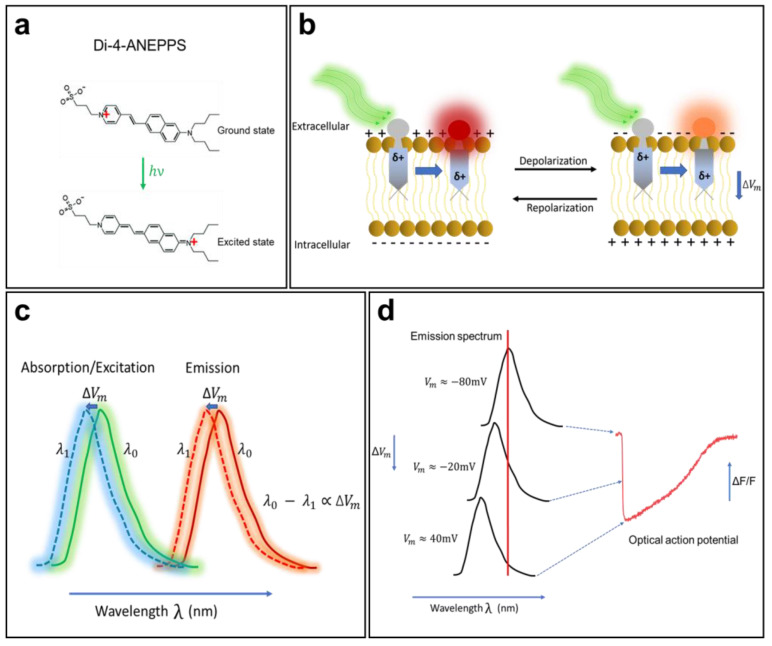
**The mechanism of voltage-sensitive electrochromic dyes**. (**a**) Voltage-sensitive dyes (VSDs) adhere to the membrane and orient so that their changing dipole is parallel to the transmembrane electric field. VSDs detect changes in surrounding potential when the generated electric field directly interacts with the chromophore. (**b**) The unexcited di-4-ANEPPS (ground state) undergoes a large dipole moment change in the presence of the electric field (excited state). (**c**) Changes in fluorophore energy levels result in small spectral changes in dye emission. (**d**) The fluorescence intensity of the probe changes in correlation with the change in membrane potential. The shift in emission spectrum is exaggerated for illustrative purpose. The AP trace in (**d**) is from unpublished data from Dr. M. Chahine’s laboratory.

**Figure 3 cells-12-02168-f003:**
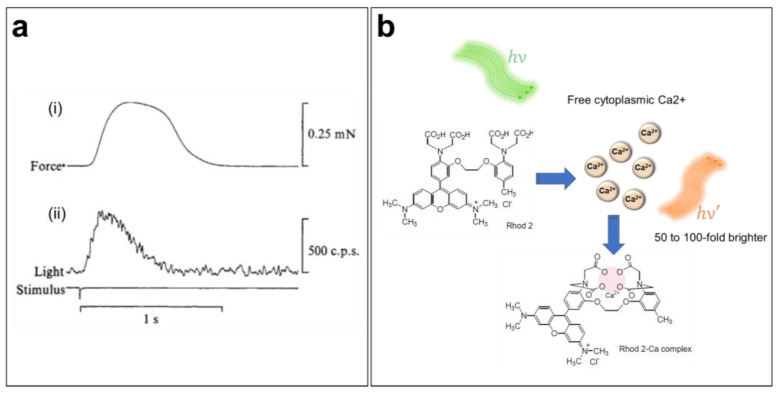
**Ca^2+^ transient imaging using fluorescent indicators**. (**a**) Mechanical (i) and luminescent (ii) responses during a contraction of a frog atrium. The first luminescence signal of a Ca^2+^ transient was obtained by injecting the cardiac muscle with aequorin, a Ca^2+^-sensitive photoprotein [61]. (**b**) Ca^2+^-sensing mechanism of the Rhod-2 AM indicator when loaded into the cell.

**Figure 4 cells-12-02168-f004:**
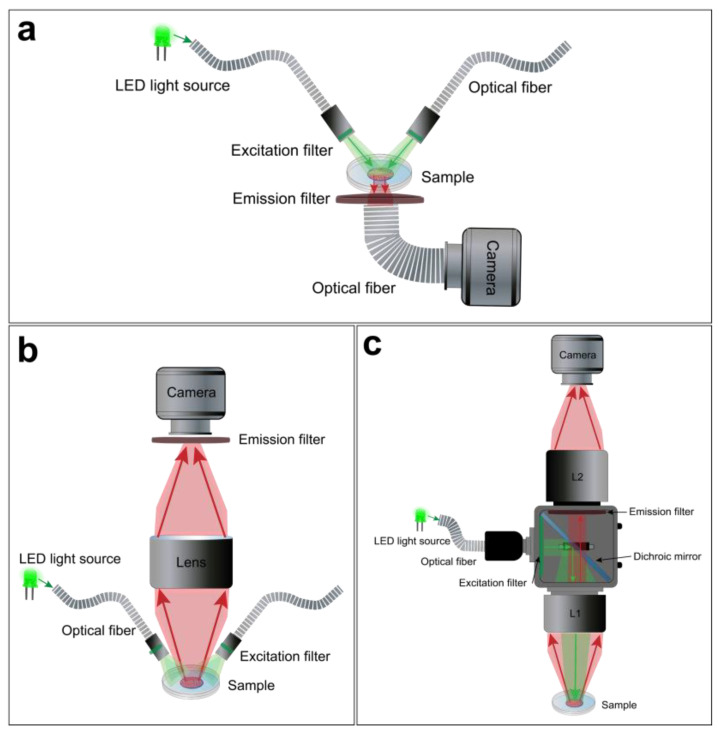
**Macroscope configurations for hiPSC-CM monolayer optical mapping**. Examples of a lensless (**a**), single lens (**b**), and tandem-lens (**c**) optical setups.

**Figure 5 cells-12-02168-f005:**
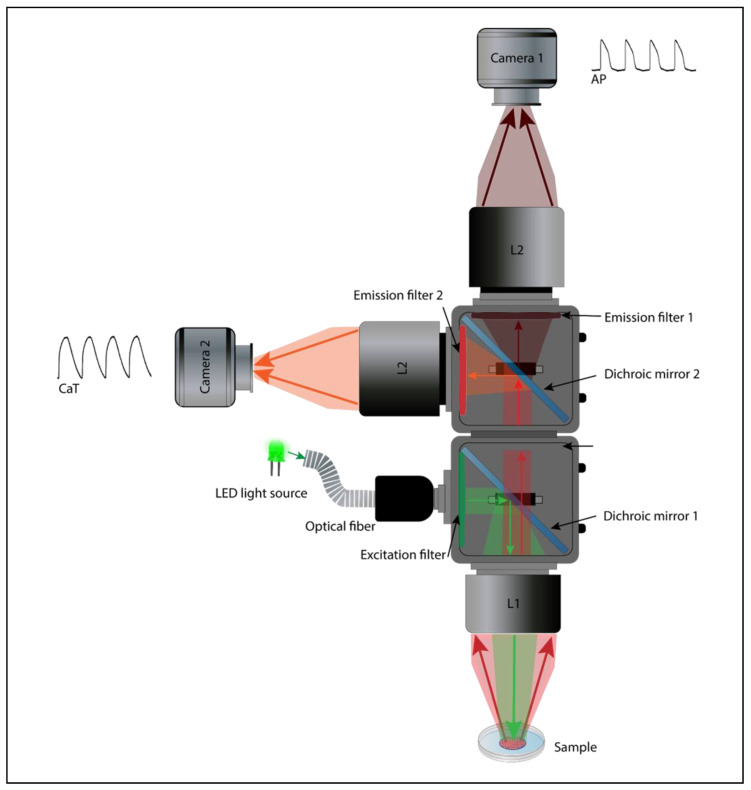
**Dual sensor optical mapping setup.** An example of a dual camera macroscope for simultaneous optical mapping of APs and CaTs in hiPSC-CM monolayers using RH237/Rhod2-AM probes combination.

**Figure 6 cells-12-02168-f006:**
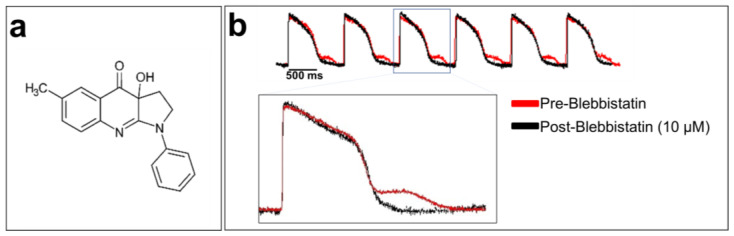
**Blebbistatin, a cardiomyocyte contraction inhibitor**: (**a**) chemical structure of blebbistatin; (**b**) optically recorded APs of hiPSC-CM monolayers loaded with di-4-ANEPPS and without blebbistatin (in red at the top of the panel). An artifact can be seen after each repolarization due to cardiomyocyte contraction. The APs in black were recorded using blebbistatin. A zoom-in of overlapped AP traces with and without blebbistatin is shown at the bottom of the figure. The traces are unpublished data from Dr. M. Chahine’s laboratory provided for illustrative purposes.

**Figure 7 cells-12-02168-f007:**
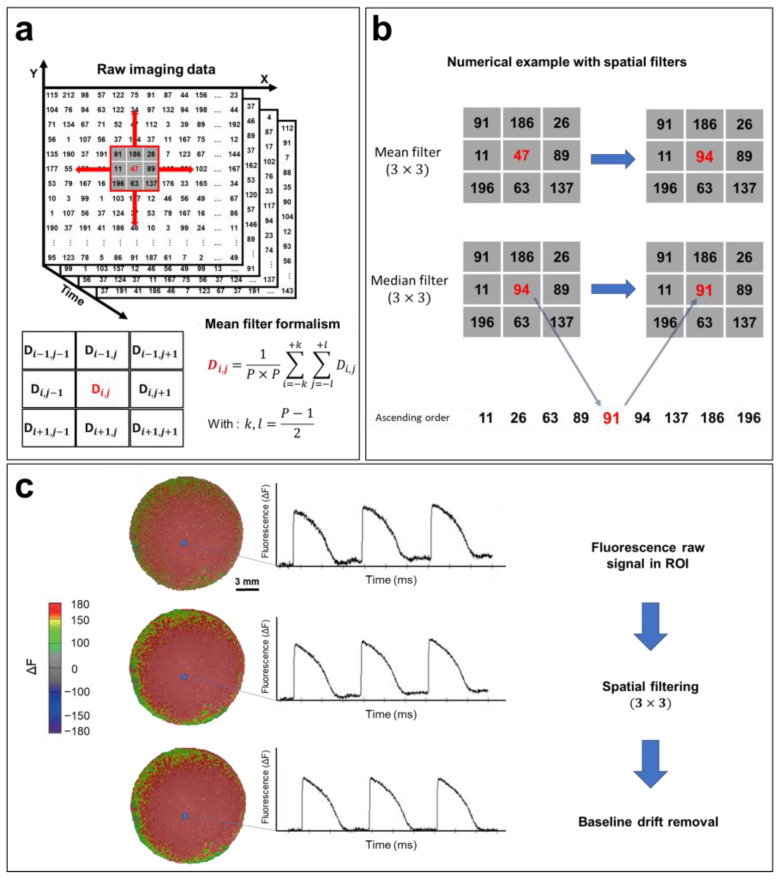
**Optical signal processing and spatial filtering**. (**a**) Representative example of optical mapping raw data (square detector), where the fluorescence intensity of each pixel is converted to a digital value in *N* matrices of (*n* × *n*), where *N* is the number of frames recorded (*N* = FPS (s^−1^) × the recording duration (s)). Fluorescence data are filtered using a mean filter formalism (bottom left). (b) Mean and median filter operations demonstrated by a numerical example. (*3* × *3*) spatial filters are applied to each pixel to remove high-frequency noise from the APs. (**c**) Processing of the optical signal. Recorded fluorescence intensity data shown as a heat map. APs are plotted from a selected ROI (blue box). Filters improve the quality of the optical signal by removing high-frequency noise, as shown in the AP traces, which also results in a smoother fluorescence intensity map. The AP baseline usually drifts because of photobleaching. By fitting this baseline drift with a polynomial or exponential equation, the drift is easily subtracted, and the baseline is adjusted closer to zero. The same steps are followed in order to process CaT optical recordings. The traces and images are unpublished data from Dr. M. Chahine’s laboratory provided for illustrative purposes.

**Figure 8 cells-12-02168-f008:**
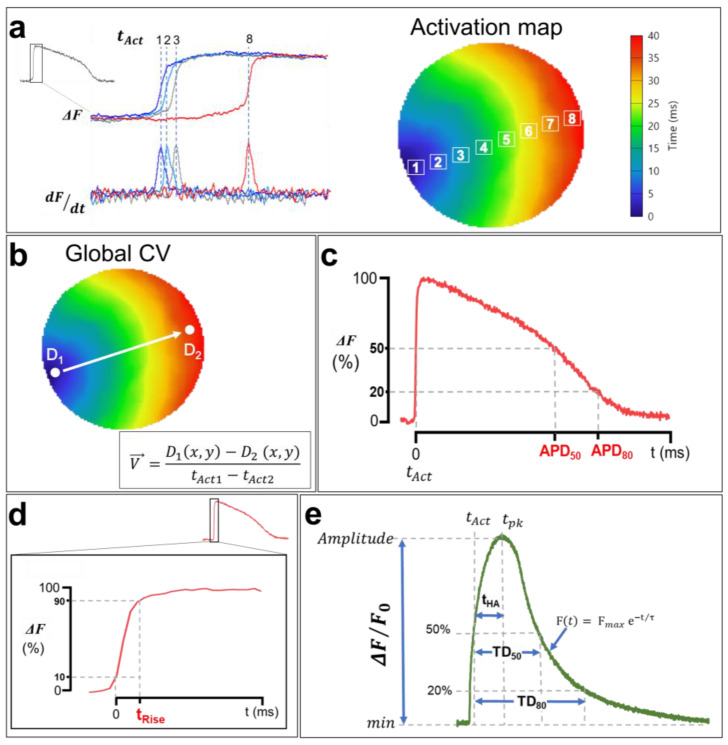
**Analysis of membrane voltage optical mapping data**. (**a**) Isochronal activation maps are generated using activation times (t_Act_) of each pixel. APs t_Act_ are measured at the maximum first derivative (dF/dt_max_), whereas t_Act_ of CaTs are measured at 50% of the upstroke phase, as shown in (**e**). Each color in the activation map corresponds to an activation time interval. (**b**) Macroscopic CV and CaPV calculation. The AP and CaT propagation rates are measured between the points D_1_ and D_2_, which were selected in the direction of the wave propagation along a path orthogonal to the propagation isochrones using the simple equation shown in the bottom right. (**c**) Optical APD measurements. APD_50_ and APD_80_ correspond to the duration from t_Act_ to the time at 50% and 80% of AP repolarization. (**d**) The rising time of AP. t_Rise_ was measured from 10% to 90% of the upstroke phase to avoid misleading noise in the optical signal even after spatial filtering. (**e**) Illustration of a CaT with the calculation methods of the TDs at 50% and 80% of reuptake, the activation half-times between the activation time t_Act_ and the time at the peak t_pk_, as well as the decay time constants (τ). The traces and images are unpublished data from Dr. M. Chahine’s laboratory, provided for illustrative purposes. AP and CaT were optically recorded using di-4-ANEPPS and Rhod-2 AM in hiPSC-CM monolayers (control cell line) at 1 Hz pacing.

## Data Availability

Not applicable.

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
