# Peer review of "Optical Mapping of Cardiomyocytes in Monolayer Derived from Induced Pluripotent Stem Cells"

_cells, 2023, doi:10.3390/cells12172168_

Round 1

Reviewer 1 Report

The subject of this review is timely; there hasn’t been a review of optical mapping of cardiomyocyte monolayers published in recent years. The organization of topics covered in the current ms is logical and would provide a solid introduction to researchers entering this field. Inclusion of a section on the analysis of optical signals is particularly welcome because this aspect has not received sufficient attention in previous reviews, unlike the preceding section topics (theoretical and practical aspects of voltage and calcium dye use, induced pluripotent stem cell monolayer cultures, and optical mapping system configurations – all of which have been reviewed elsewhere, as the ms citations indicate). However, this last section on analysis is somewhat thin and I believe it represents a missed opportunity in that it describes a minimal generic method with hardly any references to existing primary literature. For example, the methodology of conduction velocity estimation has many ramifications e.g. relating to non-uniform conduction that are not referred to or discussed in this ms. It is not “a detailed description” as alluded to in the Abstract. At the very least, this section needs to be much better referenced to allow the reader to delve into the issues further themselves.

Conduction velocity is mentioned at various points as being important, but its physiological significance is never commented on. This should be remedied.

In the Introduction the authors state “In recent years, the cultured hiPSC derived cardiomyocyte (hiPSC-CM) monolayer has become a contemporary in vitro model for the study of anisotropic conduction and arrhythmogenesis at the tissue level.” Although monolayers can provide useful information about arrhythmogenic (esp. Torsadogenic) potential of drugs, a monolayer of cultured cells is very far from real cardiac tissue (normal or diseased). Therefore its potential to elucidate mechanisms of arrhythmia in whole organs is severely limited. To their credit, the authors later say “… it is important to note that 2D hiPSC-CM monolayers do not fully capture the complexity and physiological behavior of the heart, so they should be used in conjunction with other models to obtain a comprehensive understanding of cardiac function [90].” It is good that this limitation is expressed, but the point should be further emphasized and elaborated upon. Extrapolation of the “study [of] excitation dynamics and complex arrhythmia genesis, sustenance, and termination” (referring implicitly to re-entrant mechanisms) from monolayer to whole heart is fraught with potentially unwarranted assumptions.

Notwithstanding the comments above, the review is not particularly well written in my view. There are numerous grammatical and spelling errors, inaccurate or incorrect statements, and loose expression/poor choice of words. The list below, organized by category, is not exhaustive but I hope will be useful in the major revision that I am recommending.

Grammatical and spelling errors

Line 37: “AP and Ca2+ transient (CaT) have been measured” singular-plural mismatch

Line 39: “were limited … when it comes”

Line 45: “CI” -> “CIs”

Line 58. “have opened” -> “has opened” (prepositional phrase is not part of subject)

Line 70: “arrhythmia” -> “arrhythmias”

Line 79: “proprieties”

Line 107: “A basic knowledge … are required”

Line 157: “ANEPPs” -> “ANEPPS”

Line 207: “Tyrod’s” -> “Tyrode's”

Line 227: “wildly” -> “widely”

Line 250: “which experience a significant and rapid [Ca2+]i changes”

Line 254: “Tyrod”

Line 458: “Although” -> “However”

Line 483: Delete “Although”.

Line 674: “This” -> “These”

Inaccurate or incorrect statements

Line 88: “XXV” XXV not mentioned in Ref 27. Did you mean XVII?

Line 123: “light beam with an energy hν” The individual photons have energy hν, not the beam.

Line 125: “the photons will be absorbed” Not all the photons will be absorbed.

Line 231: “Several CIs are currently available including chemical probes which are the most widely used due to their bright signal and large binding affinity to intracellular Ca2+.” High affinity indicators are not always the best choice; high-affinity indicators will saturate at too high concentrations. They also tend to slow the apparent kinetics. You later go on to write “it is important to choose a CI with a minimum perturbation of the Ca2+ dynamics” which is true. Maybe you could usefully combine these points.

Line 233: “When perfused into the cells, these indicators only bind with free cytoplasmic Ca2+ ions” This wording implies that these indicators don’t bind with other ions in solution, which is untrue.

Line 361: “if we increase the spatial resolution of a light sensor, either by widening its active surface” If the pixel area stays the same while pixel count increases, the SNR seen in each pixel will not change.

Line 376: “the light collection efficiency of a microscope is proportional to the NA of its objective.” From [93]: “For this reason the amount of exciting light through an objective is roughly proportional to (NA)^2, and at the same time the amount of the fluorescence emission collected is also proportional to (NA)^2. The intensity observed is thus proportional to the (NA)^4.”

Line 456: “the magnification tandem-lens system depends on the focal lengths of the two lenses as well as the distance between them.” Please check this; I thought the magnification depended only on the ratio of the lenses’ focal lengths, and that the distance didn’t matter (within reasonable limits) because the light rays are parallel.

Figure 7 and associated text. There is some inaccurate terminology used here. Both 3x3 averaging and median are spatial filters. Binning entails a reduction in the number of image pixels; what you call “spatial filtering” is therefore different from binning, even though both involve averaging a number of source pixels. A reduction or elimination of spatial heterogeneity (line 585) is not always desirable as your statement implies.

Incidentally, Mironov et al. (2006) used conical rather than flat PxP filters (doi:10.1152/ajpheart.01003.2005). Gaussian filtering is another alternative. This can be adjusted more finely by specifiying the filter radius in pixels. Its use adaptively is described in Pollnow et al. (2018) doi:10.1016/j.compbiomed.2018.05.029.

Line 655: “decay rate constant” -> “decay time constant”

Line 674: “[These] two-dimensional, rather than unicellular, models reproduce the same native organization of cardiomyocytes within the heart.” With respect, they don’t! Native organization is three-dimensional and therefore mechanisms like re-entry are likely to be different quantitatively and likely in more fundamental ways too.

Loose expression and poor choice of words

Line 18: “and, as such, cardiac arrhythmia mechanisms.”

Line 34: “The intracellular Ca2+ homeostasis, including Ca2+ uptake and release by sarcoplasmic reticulum and transsarcolemma influx, regulates the excitation-contraction coupling in the myocardium.” Homeostasis does not regulate EC coupling. Also, trans-sarcolemmal influx/efflux.

Line 39: “While these techniques … these standard techniques”

Line 45: “non-invasive”

Line 46: “including several colors of CIs” Color is not the important thing here; and why single out CIs when there are “several colors” of VSDs too? Sentences from lines 45-51 need to be tidied up.

Line 53: “brought a unique value” What unique value?

Line 54: “nearly replaced organic probes in several applications” Which applications?

Line 72: “when Carl J. Wiggers realized the benefits of exploiting the properties of light in cardiac electrophysiological studies, he used the same cinematographic apparatus as Mines” What properties of light? Better simply to write “Wiggers used the same cinematographic apparatus as Mines …”.

Line 90: “to obtain an optimal signal with the least noise possible” Redundant.

Line 104: “fluorophores used to record” -> “that have been used to record” (I don’t think merocyanine is used much today!)

Line 115: Sentence is long and awkward.

Line 141: “VSDs illustrate APs” Transduce?

Line 158. “are the most popular and among the most widely used” What's the difference between popular and widely used? Redundant.

Line 163: “yield” -> “follow”

Section 2.2.1 is quite repetitive

Line 206: “wider” -> “greater”

Line 207: Delete “Thus,” from “Thus, a longer emission wavelength reduces”.

Line 256: Delete “the” from “in the brain”.

Line 261: “Transient Ca2+” -> “Ca2+ transient”"

Line 262: “fluorescence contraction response” is inappropriate terminology.

Line 270: “hence their popularity”

Line 270: “another alternative was needed to circumvent the limitations as well as the problems related to the use and handling of ESCs” What limitations and what problems?

Line 284: This paragraph would benefit from a rewrite to improve clarity.

Line 302: “which are associated to the heart development” -> “which are associated with heart development”

Line 303: “Several differentiation protocols have been developed since the introduction of hiPSCs, among which, the monolayer-based differentiation as an efficient strategy leading to a highly pure population of CMs.”

Line 315: Delete “the”.

Line 321: “complex” -> “wide” or “broad” or “diverse”.

Line 333: “Thus, hiPSC-CMs provide a valuable tool for investigating and understanding the complexities of cardiac disease and high-throughput cardiotoxicity screening.” I don’t think you really mean understanding the complexities of high-throughput screening (or investigating high-thoughput screening).

Line 349: “continually used”

Line 363: “automatically” reads oddly – delete.

Line 367: “will not have enough time to collect maximum photons” Awkward wording. “will be less time to collect the same number of photons” is better.

Line 384: Delete “Indeed,”.

Line 396: “The photodetector is considered as the centerpiece of each optical mapping setup.”

Line 397: “Therefore, choice appropriate detector”

Line 398: “based on all parameters involved in experimentation”

Line 403: “pixel size must be ordered of few tens microns” ? The whole paragraph reads as a bit muddled.

Line 407: “have been detector of choice”

Line 414: “cost, knowing that price gap between CCDs and CMOS tends to decrease a lot in the recent years”

Line 439: “such as the need for transillumination mode of excitation”

Line 466: “but it is only one part of the equation.” Find a less colloquial alternative.

Line 468: “fully comprehend”

Line 472: “for further perception” There is a lot of odd wording in this section.

Line 478: “reemitted” -> “emitted”

Line 489: “which heavily impact the temporal resolution” You could say something like “reducing the temporal resolution in proportion to the number of wavelength channels”.

Line 492: “excited simultaneously by either one or two distinct wavelengths”

Line 511: “fresh culture media changing of the cardiac tissues”

Line 512: “for better and more consistent results” If better because more consistent, delete “better”; otherwise say why better.

Line 523: “to normalize the data for an adequate fixed rate comparison.” Please more precise and explicit about what you mean here.

Line 528: “Their beats” -> “Their beating” or “Their contraction”

Line 556: “mis-adaptation” ?

Line 606: “precious information” !

Line 606: “Isochrone maps provide … information about cardiac depolarization heterogeneities and intracellular Ca2+ in the hiPSC-CM monolayer.” Presumably you mean heterogeneities of cardiac depolarization and intracellular Ca2+.

Line 609: “The cell-to-cell conduction of electrical impulses and Ca2+ signaling in the myocardium depends on the ability to efficiently transfer ions between individual myocytes via gap junctions” -> “Cell-to-cell conduction of APs in the myocardium depends on the ease of the movement of ions between myocytes via gap junctions.” (Ca2+ signalling is a separate issue/process.)

Line 649: “given that channels” -> “given that Na channels”

Line 654: “CaT uptake phase” -> “the CaT uptake phase”

Line 669: “In the recent years” -> “In recent years”

Other comments, questions and suggestions

Line 38. “Patch-Clamp” -> “patch clamp”

Line 75. “Interestingly, cardiac electrophysiologists today still rely on the advantages of the optical measurements developed by Mines and Wiggers and use cinematography in high spatial resolution studies.” Ref needed.

Line 96. What is meant by “lifetime”?

Line 158. Comment on the current popularity of FluoVolt (replacing the previous ANEPPS dyes in many instances of hiPSC-CM monolayer work.

Line 183. “the imaging solution”" What does this mean?

Line 256. Italicize “in situ”.

Line 355: “allowing the propagation of a short and rapid electrical impulse followed by a Ca2+ wave propagation.” Not sure what distinction between AP and Ca events is being made here. The time-course of the depolarization is not hugely different from the Ca transient in human cells, and the delay is short and constant.

Line 411: “ultra-fast” -> “fast” and maybe quote time constant to give a quantitative idea of kinetics.

Line 418: “lasers are rather intended for imaging systems using PMT or PDA because of their very high irradiance.” Unclear what point is being made here.

Line 422: Include point about LEDs being rapidly switchable here.

Line 438: Define WD.

Line 502: “offers a significant reduction of photobleaching and phototoxicity, increased SNR, and a brighter fluorescent signal” Please explain why.

Line 511: “maturation” -> “cell maturation”

Line 515: Subscript O2 and CO2.

Line 526: “a platinum/iridium” -> “platinum or platinum-iridium”

Line 527: Lowercase “channelrhodopsins”.

Line 547: Image data is typically square in this application, but not always (and doesn’t have to be for subsequent analysis).

Line 549: “Each of these matrices is saved at a specific time according to the sampling speed.” Not really necessary to say this.

Line 553: “The background image is usually obtained by averaging the fluorescence intensity of the first four frames post-recording.” It may be, but this is not always the case. Often, a background image (or sequence of images) is recorded separately. This avoids the problem you mention in line 556.

Line 586: “vibrations”. It is unlikely that vibrations of typical timescales would be easily removable using polynomial fitting. Furthermore, exponential artefacts such as dye bleaching are better removed by exponential fitting. Being able to vary polynomial degree does not guarantee a flat baseline, let alone one that is “perfectly flat”.

Line 600: “the maximum variation of the signal relative to time (dF/dtmax)” Strictly speaking it is the point of maximum rate of increase in the signal that is used for activation time; “variation” implies it could be negative as well as positive.

Line 611: “These AP and CaT waves propagate at a given rate. These are often referred to as the CV and Ca2+ wave propagation velocity (CaPV), which are very important electrophysiological parameters.” Under what circumstances do CV and CaPV differ? That would imply a systematic change in the delay between AP upstroke and Ca transient initiation.

Line 641: “with the different calculation methods” They are not really different methods.

Line 645: “To avoid noise in the optical signal”. I think I know what you mean here, but it is unclear. When it comes to upstroke, the shape of the take-off can make it hard to estimate the start time of the upstroke, even though the baseline and peak levels can be measured accurately. Therefore, 10% and 90% levels are conventionally used. This works well because the signal is usually changing rapidly at these points, so the error in time estimation is small. (A noisy signal also makes it hard to accurately estimate when a slowly changing signal crosses a %amplitude threshold. This is why APD80 and/or TD80 is frequently preferred to APD90/TD90.)

Line 686: The contribution of the second MC author should be included.

Figures

A general criticism of the figures is that the source of original traces/images are not always given. Figure 1 is okay, but Figures 6, 7 and 8 are missing references.

Figure 2-b. Rather than showing the shorter wavelength emission as blue, orange would be closer to the truth and still indicate a difference visually.

Figure 2-c is confusing/misleading. The blue leftward pointing arrow needs to go under the pairs of spectra, not between pairs.

Figure 2-d. Line 199. It would be helpful to say the shift is hugely exaggerated. What is shown is inconsistent with text “molecule undergoes a shift of a few nanometers to a shorter wavelength” (assuming a normally broad emission spectrum).

Figure 3 citation is incorrect. [32] does not contain an aequorin trace.

Figure 3-b. Why are the calcium ions “glowing”?

Figure 7-c. Color scale in key appears to be different from the three images to the left.

Figure 8-a. How was this obtained? Stimulation method/protocol needs to be specified. For dF/dt to have its conventional meaning (i.e. rate of change of fluorescence), the top trace should be labelled F (or possibly DF for consistency with panels c, d & e).

Figure 8-b (line 616). How are the two groups of pixels at D1 and D2 selected? Presumably they have some relation to early and late activation. What is denoted by D1(x,y) and D2(x,y), and what are the units? Given the important measure is the distance between the groups, isn’t there a simpler and less potentially confusing way to express this on the diagram.

References

When citing references, it is better to retain the original capitalization of titles, rather than Capitalizing Each Word. However, I will defer to the journal format if that differs.

Himel et al. (2012) review “Optical imaging of arrhythmias in the cardiomyocyte monolayer” in Heart Rhythm not cited.

Tung & Cysyk (2007) also relevant but not cited.

Swift et al. (2021) article in AJP might also be included as a reference relevant to blebbistatin use.

Consider removing Ref 36. There are better choices that a Scholarpedia article, and in this instance not much value is added by its inclusion.

Ref 80 does not have DOI. (It is doi: 10.1016/j.ceca.2016.02.004)

Ref 103 (cited in line 477) does not have DOI and appears to be wrong. Efimov et al. paper is about repolarization and refractoriness. Did you mean Choi & Salama (2000) Ref [104]?

A good reference for “Processing and analysis of cardiac optical mapping data obtained with potentiometric dyes” is Laughner et al. (2012) doi: 10.1152/ajpheart.00404.2012

See comments and suggestions for Authors.

Reviewer 2 Report

The manuscript offers a detailed and technical description of the optical mapping technique for most of its part.

Comments, major and minor:

Line 268: the abbreviation hESCs should be defined here

Section 4

- title: hiPSC-CM would be more accurate (instead of iPSC-CM)

- an introductory sketch on the particularities and  challenges of performing optical mapping in iPSC-CM monolayers would be welcomed

- lines 349 – 351 – the information is redundant with the previous subsections

Section 4 is too descriptive and technical in most of its content, a detailed presentation of an assay. Instead, would be expected to be a critical presentation of a cutting edge technique, with its advantages,  challenges, latest advancements and limitations underlined, as evidenced in the literature; when the direct experience of the authors is the source, the fact is expected to be clearly announced. Also, the information should be focused on the particularities of optical mapping on hiPSC-CM, as announced by the titles of the manuscript and section 4.

Page 15, first paragraph, and figure legend 7: appropriate references should be provided. The same applies to subchapter 4.5 and figure legend 8.

Would be highly recommended to mention recent and potential applications of the method, with examples, and its limitations - with a constructive leap towards future developments.

Conclusions – too general and elusive 

Round 2

Reviewer 1 Report

The revised ms reads much better now. I believe it will be a useful review for researchers interested in the field of mapping cardiomyocyte monolayers.

Reviewer 2 Report

Thank you for accurately addressing all comments